

# Brain tumor segmentation using U-Net in conjunction with EfficientNet

Shu-You Lin and Chun-Ling Lin

Department of Electrical Engineering, Ming Chi University of Technology, New Taipei City, Taiwan

## ABSTRACT

According to the Ten Leading Causes of Death Statistics Report by the Ministry of Health and Welfare in 2021, cancer ranks as the leading cause of mortality. Among them, pleomorphic glioblastoma is a common type of brain cancer. Brain cancer often occurs in the brain with unclear boundaries from normal brain tissue, necessitating assistance from experienced doctors to distinguish brain tumors before surgical resection to avoid damaging critical neural structures. In recent years, with the advancement of deep learning (DL) technology, artificial intelligence (AI) plays a vital role in disease diagnosis, especially in the field of image segmentation. This technology can aid doctors in locating and measuring brain tumors, while significantly reducing manpower and time costs. Currently, U-Net is one of the primary image segmentation techniques. It utilizes skip connections to combine high-level and low-level feature information, leading to significant improvements in segmentation accuracy. To further enhance the model's performance, this study explores the feasibility of using EfficientNetV2 as an encoder in combination with U-net. Experimental results indicate that employing EfficientNetV2 as an encoder together with U-net can improve the segmentation model's Dice score (loss = 0.0866, accuracy = 0.9977, and Dice similarity coefficient (DSC) = 0.9133).

# INTRODUCTION

According to the Ten Leading Causes of Death Statistics Report by the Ministry of Health and Welfare in 2021 (*Chiu et al., 2021*), cancer ranks as the leading cause of death nationwide. Among them, pleomorphic glioblastoma, also known as multiple morphologic glioblastoma, is a common brain cancer. This tumor can be categorized into lower-grade glioma and higher-grade glioma. Lower-grade gliomas have a lower risk, whereas higher-grade gliomas are malignant tumors, characterized by fast growth and invasive properties.

Currently, the diagnosis of brain tumors heavily relies on magnetic resonance imaging (MRI), providing crucial brain images for analysis. Nevertheless, the process of manually locating and measuring these tumors demands a high level of expertise from doctors and significantly consumes both human resources and time. This presents a significant challenge in the healthcare system. Therefore, automated or semi-automated methods are crucial to facilitate locating and measuring the tumors. Several methods have been

Corresponding author
Chun-Ling Lin,
ginnylin@mail.mcut.edu.tw

proposed for tumor segmentation (*Aboelenein et al., 2020*; *Zhang et al., 2020*), which utilize advanced image segmentation and deep learning techniques. These methodologies significantly reduce the labor-intensive aspects of manual segmentation and analysis, saving considerable time and effort for medical professionals. By effectively addressing these challenges, these methods not only enhance diagnostic accuracy but also expedite the process, allowing doctors to redirect their focus toward patient treatment and care, thus amplifying the efficiency of healthcare processes.

Image segmentation plays a crucial role in various medical applications, allowing for the precise identification and delineation of structures in medical images. In this context, U-Net (*Ronneberger, Fischer & Brox, 2015*) stands as a noteworthy deep learning network architecture introduced by *Ronneberger, Fischer & Brox (2015)*. Its significance lies in its specialization for image segmentation tasks and its capability to achieve accurate and efficient segmentation. The name "U-Net" is derived from its characteristic U-shaped architecture, composed of two key components: the front-end encoder and the back-end decoder. The encoder's role is to down-sample the input image and extract essential features using multiple convolutional and pooling layers. This process captures high-level features while progressively reducing the image size. On the other hand, the decoder is responsible for up-sampling and restoring the image to its original size, aiming to recover fine details and local information. What makes U-Net particularly powerful is its use of skip connections, which connect encoder feature maps to their corresponding maps in the decoder. These connections preserve detailed information and significantly enhance the model's segmentation performance.

In summary, U-Net is well-suited for medical image segmentation (*Du et al., 2020*; *Siddique et al., 2021*), and its encoder, decoder, and skip connections enable accurate image segmentation, leading to successful results in various medical image segmentation tasks. However, it is important to note that while U-Net is a widely utilized and powerful model for image segmentation, it does have certain limitations (*Hu & Yang, 2020*; *Wei et al., 2021*; *Woo & Lee, 2021*). One notable concern is its susceptibility to overfitting when dealing with limited sample sizes, resulting in reduced generalization ability when applied to new test data. Moreover, in segmentation tasks where 'class imbalance' refers to a situation where certain classes have significantly fewer instances than others, U-Net may struggle to adequately segment these underrepresented classes, leading to potential impacts on the overall segmentation accuracy. Furthermore, due to its deep architecture, U-Net tends to yield larger model sizes which are computationally expansive, which can present challenges in scenarios with limited computational resources. To address these limitations, an alternative approach involves the introduction of EfficientNet (*Abedalla et al., 2021*; *Wang et al., 2021*) as an encoder in the image segmentation model. EfficientNet (*Koonce & Koonce, 2021*) is recognized for its efficient network structure and exceptional feature extraction capabilities, allowing it to achieve comparable accuracy to larger models while maintaining a smaller model size and computational cost.

*Wang et al. (2021)* introduced an enhanced method based on the U-Net framework, which leverages EfficientNetB4 as the encoder to facilitate more comprehensive feature extraction during the encoding stage. They also introduced an attention gate in the skip

connection to selectively focus on relevant features for the specific segmentation task. To address the challenge of gradient vanishing, the authors replaced the conventional decoder convolution with a residual block, leading to a notable improvement in segmentation accuracy. In the Sliver07 evaluation, the proposed method demonstrates superior segmentation performance across all five-standard metrics. Similarly, in the LiTS17 assessment, it achieves the best performance, with only a slight difference in RVD. Furthermore, their participation in the MICCAI-LiTS17 challenge resulted in an impressive Dice per case score of 0.952.

*Abedalla et al. (2021)* presented a novel end-to-end semantic segmentation model named Ens4B-UNet for medical images. This model combined four U-Net architectures with pre-trained backbone networks, leveraging powerful convolutional neural networks (CNNs) as backbones for U-Net's encoders and utilizing nearest-neighbor up-sampling in the decoders. Ens4B-UNet was designed based on the weighted average ensemble of four encoder–decoder segmentation models. All the backbone networks of the ensembled models were pre-trained on the ImageNet dataset to exploit the benefits of transfer learning. To improve their models, the authors applied several techniques for training and predicting, including stochastic weight averaging (SWA), data augmentation, test-time augmentation (TTA), and different types of optimal thresholds. They evaluated and tested their models on the 2019 Pneumothorax Challenge dataset, which contained 12,047 training images with 12,954 masks and 3,205 test images. The proposed segmentation network achieved a mean Dice similarity coefficient (DSC) of 0.8608 on the test set, ranking among the top one-percent systems in the Kaggle competition.

By integrating EfficientNet as an encoder, the image segmentation model gains the capacity to capture crucial features of brain tumors, leading to improved segmentation accuracy and efficiency. Furthermore, EfficientNet's feature extraction ability plays a vital role in mitigating overfitting concerns, enhancing the model's generalization to new test data, and facilitating its application in diverse real-world scenarios. Additionally, this feature extraction capability helps tackle class imbalance issues, ensuring appropriate segmentation for each class. In conclusion, the integration of EfficientNet as an encoder effectively addresses U-Net's limitations and significantly enhances the performance of the image segmentation model in brain tumor segmentation tasks. This advancement contributes to a more efficient and robust solution for medical image segmentation.

EfficientNetV2 (*Tan & Le, 2021*), developed by Google in 2021, is an optimized convolutional neural network architecture that builds upon the original EfficientNet. It combines the strengths of MBConv and Fused-MBConv to enhance performance while managing parameters efficiently, achieving a balance between computational efficiency and accuracy. As a result, the primary objective of this study is to explore the incorporation of EfficientNetV2 as an encoder in brain tumor segmentation, leveraging its superior feature extraction capabilities to enhance the performance of the image segmentation model while maintaining a relatively smaller model size and computational cost, achieving exceptional accuracy. Compared to traditional manual segmentation methods, the approach of combining EfficientNetV2 as an encoder with U-net enables automatic identification and segmentation of brain tumor regions, reducing the workload for medical professionals.

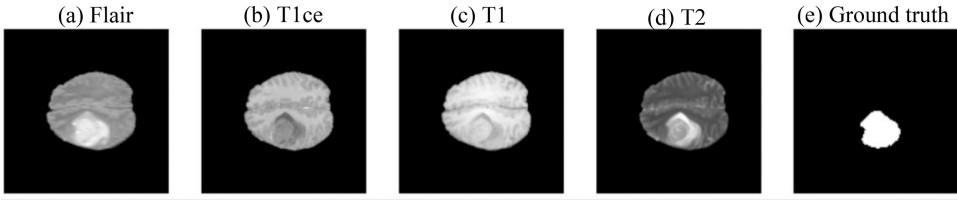

**Figure 1** **Example of magnetic resonance imaging (MRI) data (*Bakas et al., 2017a*; *Bakas et al., 2017b*; *Bakas et al., 2018*; *Clark et al., 2013*; *Menze et al., 2014*), CC0: Public Domain).** (A) FLAIR, (B) T1ce, (C) T1, (D) T2, and (E) Ground truth.

Furthermore, this method improves segmentation accuracy and efficiency, facilitating quicker diagnosis and formulation of appropriate treatment plans. The main focus of this research is to evaluate the performance of the EfficientNetV2 encoder combined with U-net in brain tumor image segmentation and investigate its impact on model accuracy and efficiency. Through this approach, the aim is to achieve superior results in brain tumor segmentation, enabling early detection and diagnosis of malignant brain tumors like pleomorphic glioblastoma. This, in turn, will enhance treatment accuracy, reduce patient risks, and provide medical professionals with advanced tools for brain tumor image analysis.

## METHODS

The 'Methods' section of this study encompasses the specific procedures employed to achieve image standardization of brain magnetic resonance images (MRI) through preprocessing. The study will first explore various preprocessing methods ensuring consistency in image data, including adjustments for size and resolution. Subsequently, the approach for arranging and combining preprocessed images as input for model training will be detailed. Lastly, the implementation of a majority voting mechanism to combine trained models for an automated brain tumor segmentation system will be elucidated. Together, these sequential steps offer a comprehensive solution to augment the efficiency of brain MRI processing and analysis.

### Datasets

This study utilized the Brats 2019 dataset provided by the Medical Image Computing and Computer Assisted Intervention Society (MICCAI) (*Bakas et al., 2017a*; *Bakas et al., 2017b*; *Bakas et al., 2018*; *Clark et al., 2013*; *Menze et al., 2014*) (https://www.kaggle.com/datasets/aryashah2k/brain-tumor-segmentation-brats-2019/data, CC0: Public Domain) for training, validation, and testing of the model. The dataset comprises 335 cases, each containing four types of magnetic resonance imaging (MRI) modalities (each MRI image size is $240 \times 240 \times 155$, as shown in Fig. 1), including FLAIR, T1ce, T1, and T2 and Ground truth.

1. Fluid-attenuated inversion recovery (FLAIR) utilizes specialized pulse sequences and inversion recovery techniques to suppress the signal from cerebrospinal fluid (CSF) and detect water content in brain tissues.

**Table 1  Area of defective MRI images.**

|  | FLAIR | T1ce | T1 | T2 |
|---|---|---|---|---|
| Area | 9245.5 | 14609 | 14617 | 14625.5 |

2. T1-weighted imaging (T1) generates high contrast images of brain structures by using the high signal from fatty tissues and low signal from fluid tissues.

3. T1-weighted post-contrast enhancement (T1ce) involves the use of a contrast agent after T1-Weighted Imaging, allowing the detection of brain hemorrhages as the contrast agent enters the brain through blood circulation.

4. T2-weighted imaging (T2) uses the high signal from fluid tissues to detect water content in brain structures.

5. Ground truth: The actual annotated regions of brain tumors.

These magnetic resonance imaging (MRI) techniques provide various information and play a crucial role in comprehensively understanding brain structures and abnormalities, contributing significantly to brain tumor segmentation tasks.

## Image prepressing

If the data specifications are not standardized, it can pose challenges during model training. Such data inconsistencies may result in a time-consuming training process as the model needs to handle images of varying resolutions. Moreover, irrelevant features might be absorbed by the trained model, leading to negative impacts on its efficiency and accuracy. Therefore, performing appropriate preprocessing is a crucial step to address this issue. By unifying the image specifications, we can ensure that the model processes consistent data during training and focuses only on essential features, thus enhancing its efficiency and performance. This study proposes four preprocessing steps for brain tumor magnetic resonance images includes defective image removal, brightness normalization, image resizing and cropping, and data normalization, as illustrated in Fig. 2. The following sections will provide detailed explanations for each preprocessing method.

### Defective image removal

Due to the operation of the scanner, there may be slight defects in the acquired MRI images. As shown in Fig. 3, the FLAIR image (Fig. 3A) is incomplete compared to the FLAIR image in Fig. 1A. These defects can significantly impact the model training and ultimately lead to a decrease in model performance. To prevent this performance degradation, we utilize OpenCV algorithms to calculate the image area and automatically filter out data with defects using the Dixon's Q-test.

First, the OpenCV algorithm's threshold() method is used to convert the image into a binary image (*Bradski, 2000*; *Bradski & Kaehler, 2000*), as shown in Figs. 3E–3H. After the binary processing, the cv2.contourArea() function is utilized to calculate the area of the image, as shown in Table 1.

Finally, Dixon's test (*Saleem, Aslam & Shaukat, 2021*) is applied to screen the data for the presence of outliers. The Dixon's test calculates the Q-statistic value of the data and confirms whether it is an outlier based on the critical value of the significance level. The

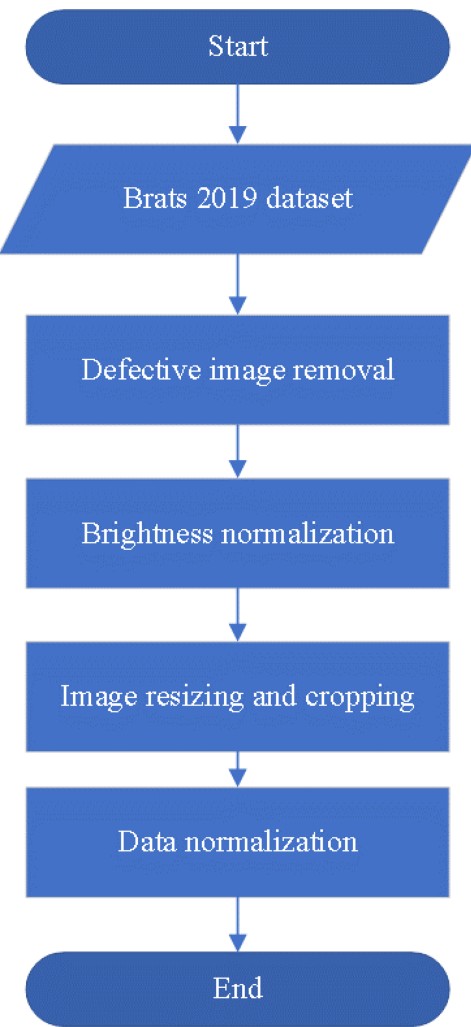

**Figure 2  The preprocessing flowchart.**

Q-statistic value is calculated using formula (1).

$$Q = \frac{x(2) - x(1)}{x(n) - x(1)} \tag{1}$$

The area data in Table 1 is sorted in ascending order, and the Q-statistic value is computed using formula (1), *i.e.*, (14609–9245)/(14625.5–9245.5) = 0.9969. Then, it is compared with the critical value in Table 2, where the significance level ($\alpha$) in this study is set to 0.001, resulting in a critical value (N) of 0.964. As the Q-statistic value is greater than the critical value, it indicates the presence of an outlier in the data, suggesting that one of the four images has a significantly lower area compared to the other three images. Due to the varying sizes of regions in the images, it may lead to a decline in model performance. Therefore, in Brats 2019, the four problematic cases will be automatically removed, leaving a total of 331 cases for further research and validation.

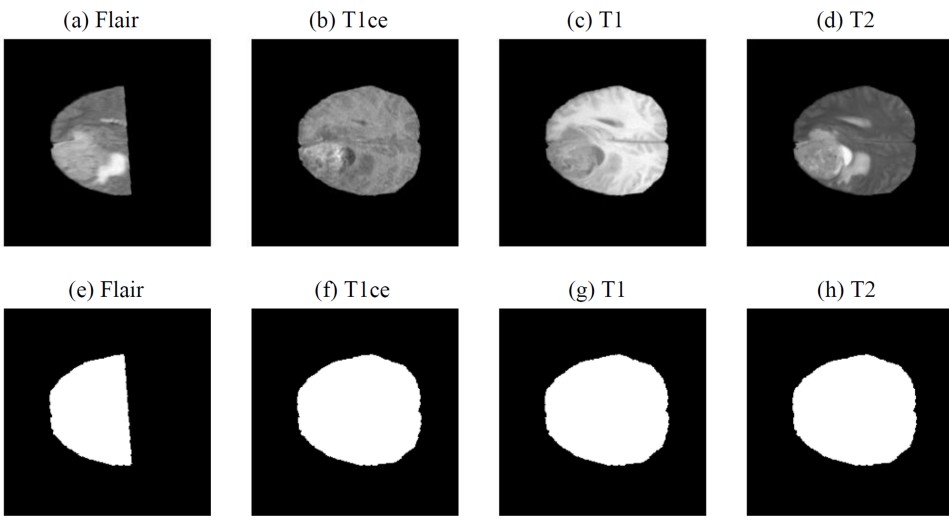

**Figure 3** **Defective MRI images** (*Bakas et al., 2017a*; *Bakas et al., 2017b*; *Bakas et al., 2018*; *Clark et al., 2013*; *Menze et al., 2014*, **CC0: Public Domain**). (A) FLAIR, (B) T1ce, (C) T1 and (D) T2 and Defective MRI images after binary processing ((E) FLAIR, (F) T1ce, (G) T1 and (H) T2.

**Table 2** **Critical value distribution of area table.**

| $\alpha$ | 0.001 | 0.002 | 0.005 | 0.01 | 0.02 | 0.05 | 0.1 | 0.2 |
|---|---|---|---|---|---|---|---|---|
| N | 0.964 | 0.949 | 0.921 | 0.889 | 0.847 | 0.766 | 0.679 | 0.561 |

**Notes.**
$\alpha$: Significance level.

### Brightness normalization

Due to the inherent magnetic field inhomogeneity of the magnetic resonance imaging (MRI) equipment, brightness non-uniformity is a common issue in MRI images. In this study, we employed the N4ITK (Non-Local Means-based Bias Field Correction) bias correction method (*Tustison et al., 2010*). This method utilizes a non-local means filter to locally smooth the images, reducing brightness non-uniformity and intensity variations, as shown in Fig. 4. Through this correction method, a more uniform brightness distribution is achieved, enhancing the comparability and interpretability of the images, and improving the accuracy of the model. This step helps eliminate biases in the imaging process, leading to a more stable and reliable training and analysis process.

### Image resizing and cropping

Due to the limited resources in the Google Colab environment used in this study, and in order to save model parameters, the $240 \times 240 \times 155$ three-dimensional data was scaled down to $128 \times 128 \times 155$. This helps to improve the training efficiency of the model within the constrained resources.

Furthermore, to utilize a two-dimensional model instead of a three-dimensional model, the study sliced the three-dimensional data into three orthogonal planes: coronal, axial, and sagittal, as shown in Fig. 5. This processing approach allows the study to handle the

(a)          (b)

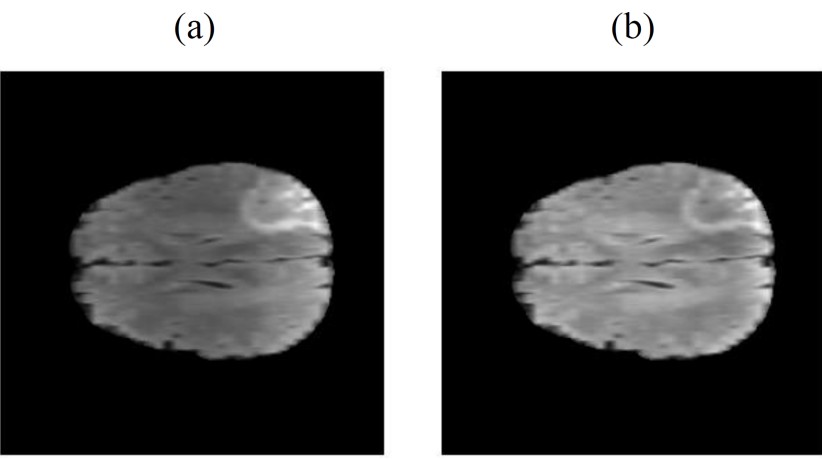

**Figure 4** **The N4ITK bias correction method is applied, where (A) represents the original uncorrected image** (*Bakas et al., 2017a*; *Bakas et al., 2017b*; *Bakas et al., 2018*; *Clark et al., 2013*; *Menze et al., 2014*, **CC0: Public Domain), and (B) represents the corrected image.**

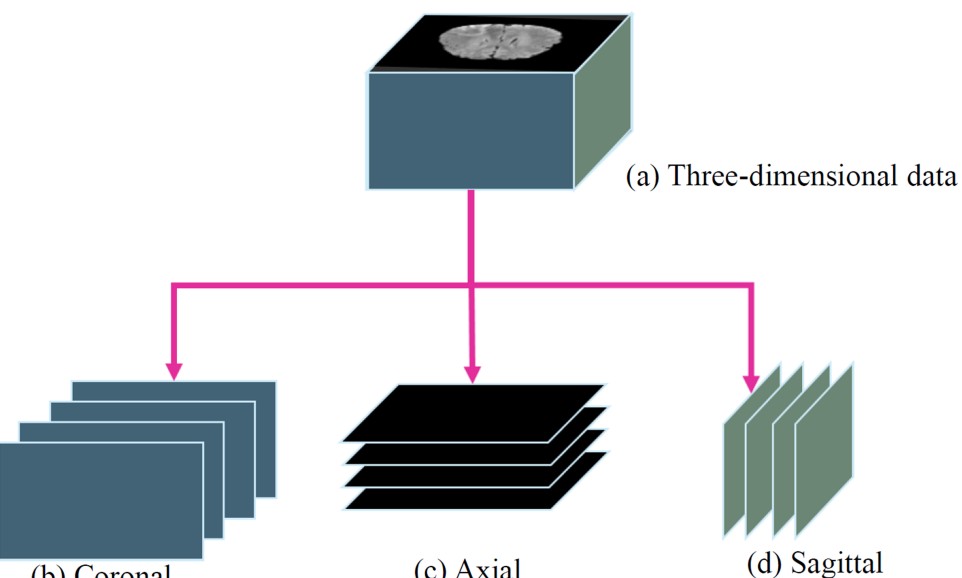

(a) Three-dimensional data

(b) Coronal          (c) Axial          (d) Sagittal

**Figure 5** **(A) Three-dimensional data into three orthogonal planes** (*Bakas et al., 2017a*; *Bakas et al., 2017b*; *Bakas et al., 2018*; *Clark et al., 2013*; *Menze et al., 2014*, **CC0: Public Domain): (B) coronal, (C) axial and (D) sagittal.**

data in a two-dimensional format, reducing the complexity and computational burden of the model. It facilitates efficient training and prediction within the limited resources available.

### Data normalization

Data normalization is a crucial step in model training as it enhances the model's stability, prevents the influence of extreme values, and accelerates the convergence speed. Without data normalization, the model may suffer from low performance and issues like gradient vanishing during training. Therefore, this study adopts Z-score normalization (*Hou et al., 2019*), as shown in Eq. (2):

$$z' = \frac{z - \mu}{\delta} \tag{2}$$

where z and z' are the input image and the normalized image, respectively. $\mu$ represents the mean value of the input image, and $\delta$ represents the standard deviation of the input image. Through this method, we transform the data distribution into a standard distribution with a mean of 0 and a standard deviation of 1. The benefit of this approach is that the model can more easily learn the features of the data during training and reduces the impact of biases in the data, thereby improving the stability and predictive capability of the model.

## Proposed brain tumor segmentation method

In medical image segmentation, the U-Net architecture, introduced by *Ronneberger, Fischer & Brox (2015)* has gained recognition. U-Net comprises an encoder and a decoder. The encoder handles down-sampling and feature extraction, while the decoder is responsible for up-sampling and feature fusion. The use of skip connections between the encoder and decoder allows U-Net to retain intricate image details. Recent improvements in U-Net include variations like ResUNet++ (*Jha et al., 2019*) for encoder enhancements and Attention UNet++ (*Li et al., 2020*) for decoder improvements.

This study references the EAR-U-Net liver tumor segmentation architecture in 2021 (*Wang et al., 2021*). This study employs EfficientUNet, a modified U-Net version that utilizes the EfficientNet architecture as its encoder. EfficientNet combines depth and feature extraction capabilities with efficiency. It is known for maintaining excellent performance while having fewer parameters and computations. To compare U-Net and EfficientUNet for Brain Tumor Segmentation, separate modules using each architecture were constructed.

### U-Net

U-Net, designed by *Ronneberger, Fischer & Brox (2015)*, is tailored for image segmentation. It consists of an encoder for down-sampling and feature extraction and a decoder for up-sampling. Skip connections connect feature maps between the encoder and decoder, preserving intricate details for better segmentation performance.

The U-Net model used here, as illustrated in Fig. 6, consists of five convolutional blocks, each with two convolutional layers. Batch normalization enhances convergence, and ReLU is the activation function. The model has a total of 7,772,161 parameters, as shown in Fig. 7.

### EfficientUNet

EfficientNet (*Tan & Le, 2019*), introduced by Google in 2019, is a highly efficient convolutional neural network architecture. It scales depth, width, and image size uniformly

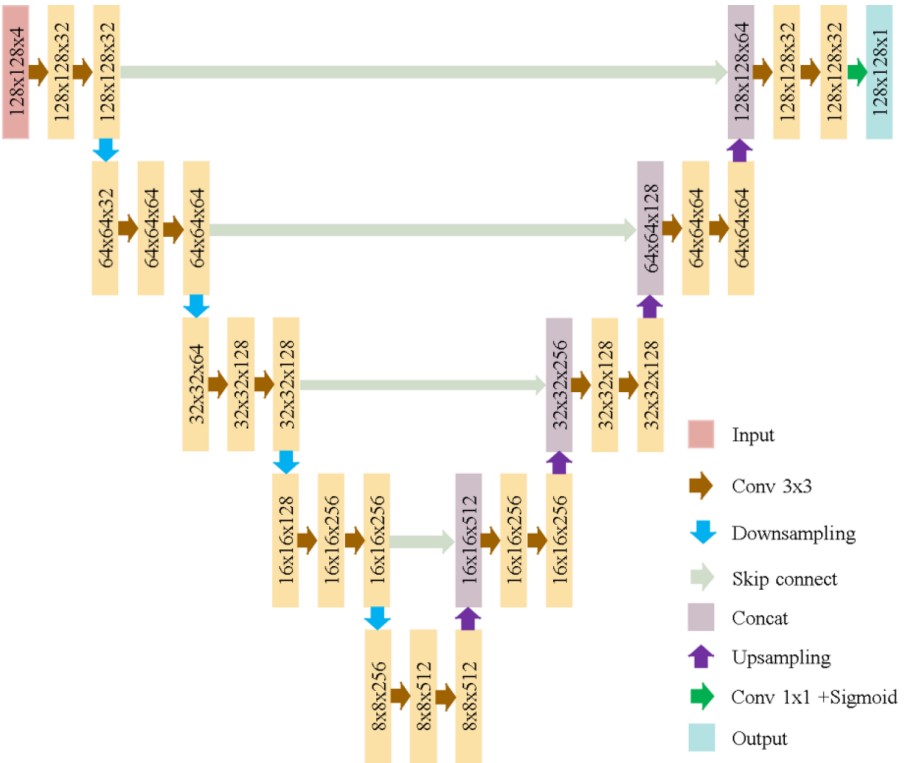

**Figure 6  U-Net architecture in this study.**

```
re_lu_20 (ReLU)                    (None, 128, 128, 32   0        ['batch_normalization_16[0][0]']
                                   )

conv2d_21 (Conv2D)                 (None, 128, 128, 32   9248     ['re_lu_20[0][0]']
                                   )

batch_normalization_17 (BatchN     (None, 128, 128, 32   128      ['conv2d_21[0][0]']
ormalization)                      )

re_lu_21 (ReLU)                    (None, 128, 128, 32   0        ['batch_normalization_17[0][0]']
                                   )

conv2d_22 (Conv2D)                 (None, 128, 128, 1)   33       ['re_lu_21[0][0]']

==================================================================================================
Total params: 7,772,161
Trainable params: 7,766,273
Non-trainable params: 5,888
```

**Figure 7  The number of parameters in U-Net.**

for efficiency. In this study, we refer to EfficientNetV2S (*Tan & Le, 2021*) due to its higher accuracy and shorter training time compared to EfficientNetB4.

EfficientUNet replaces the U-Net's encoder with EfficientNetV2S and uses a conventional convolutional decoder, as detailed in Table 3. The decoder contains five convolutional blocks, each with two convolutional layers. Batch normalization and ReLU activation are applied. The architecture of EfficientUNet is illustrated in Fig. 8. The

**Table 3  EfficientNetV2S architecture.**

| Stage | Operator | Channels |
|---|---|---|
| 0 | Conv3 ×3 | 24 |
| 1 | Fused-MBConv1, k3 ×3 | 24 |
| 2 | Fused-MBConv4, k3 × 3 | 48 |
| 3 | Fused-MBConv4, k3 × 3 | 64 |
| 4 | MBConv4, k3 × 3, SE0.25 | 128 |
| 5 | MBConv6, k3 × 3, SE0.25 | 160 |
| 6 | MBConv6, k3 ×3, SE0.25 | 256 |
| 7 | Conv 1 ×1&Pooling&FC | 1,280 |

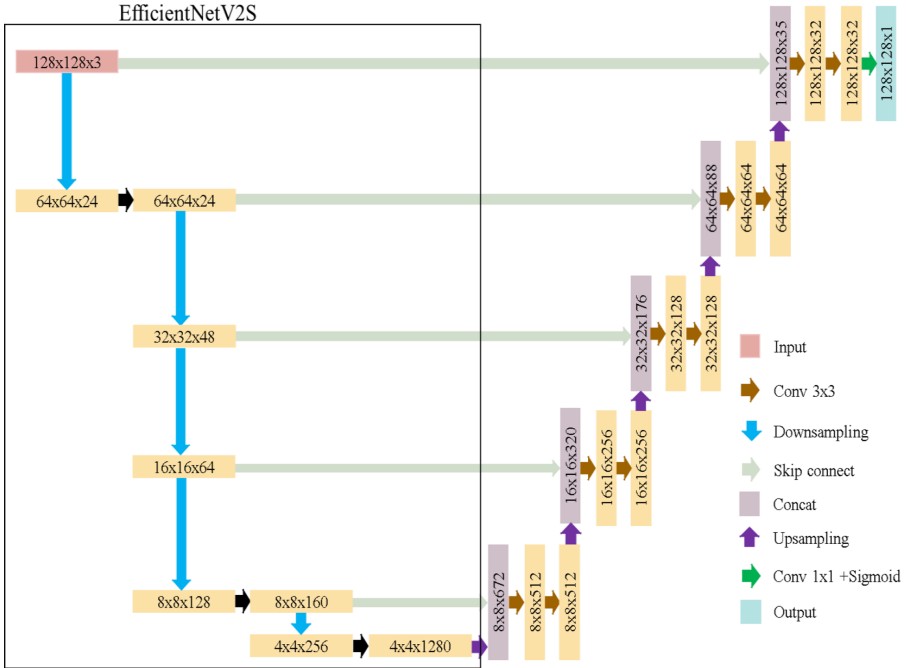

**Figure 8  EfficientUNet in this study.**

EfficientUNet model uses 30,900,097 parameters for feature extraction and segmentation, as presented in Fig. 9.

### Applying the U-Net and EfficientUNet to brain tumor segmentation prediction system

The workflow of the U-Net prediction system in this study is shown in Fig. 10. Initially, the dataset is sliced into three orthogonal two-dimensional planes: axial, sagittal, and coronal. Each plane is then fed into the U-Net for prediction separately. Finally, the results from the three planes are combined using a majority voting method (*Van Erp, Vuurpijl & Schomaker, 2002*) to obtain the final predicted segmentation map.

```
conv2d_9 (Conv2D)              (None, 128, 128, 32  9248      ['activation_8[0][0]']
                               )

batch_normalization_9 (BatchNo (None, 128, 128, 32  128       ['conv2d_9[0][0]']
rmalization)                   )

activation_9 (Activation)      (None, 128, 128, 32  0         ['batch_normalization_9[0][0]']
                               )

conv2d_10 (Conv2D)             (None, 128, 128, 1)  33        ['activation_9[0][0]']

==================================================================================================
Total params: 30,900,097
Trainable params: 30,742,257
Non-trainable params: 157,840
```

**Figure 9**  **The number of parameters in EfficientUNet.**

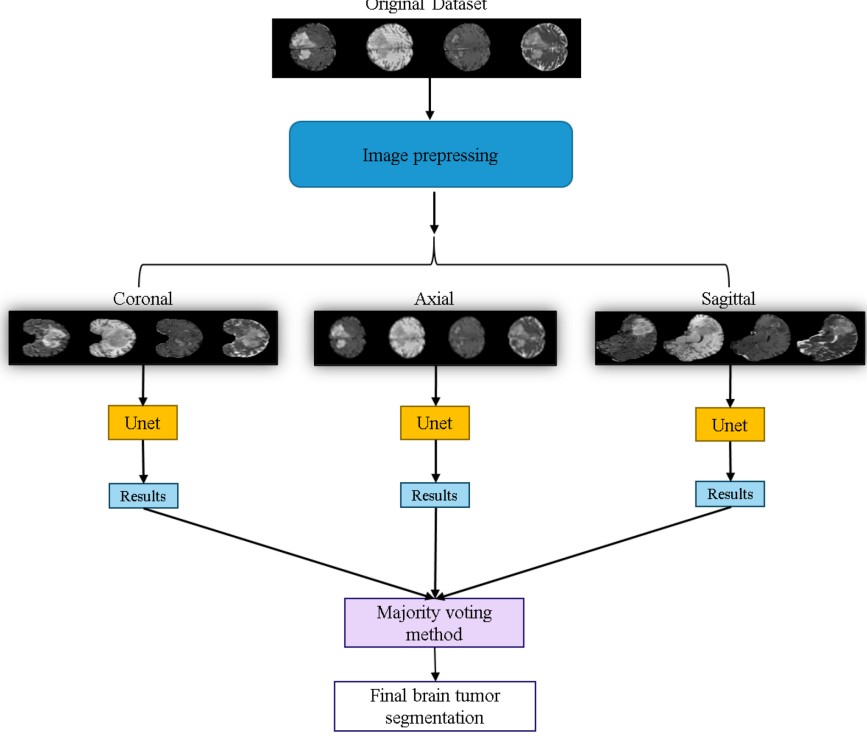

**Figure 10**  **Application of the U-Net for brain tumor segmentation workflow.** Original dataset (*Bakas et al., 2017a*; *Bakas et al., 2017b*; *Bakas et al., 2018*; *Clark et al., 2013*; *Menze et al., 2014*, CC0: Public Domain).

The workflow of the EfficientUNet prediction system in this study is illustrated in Fig. 11. Firstly, the dataset is preprocessed and sliced into three 2D data orientations, namely axial, sagittal, and coronal. Next, the original four-channel data is arranged and combined into four sets of three-channel data, which are (FLAIR, T1ce, T1), (FLAIR, T1ce, T2), (T2, T1ce, T1), and (FLAIR, T1, T2). Each of these three-channel data sets is individually input into the EfficientUNet model for prediction. For each orientation, the best prediction is

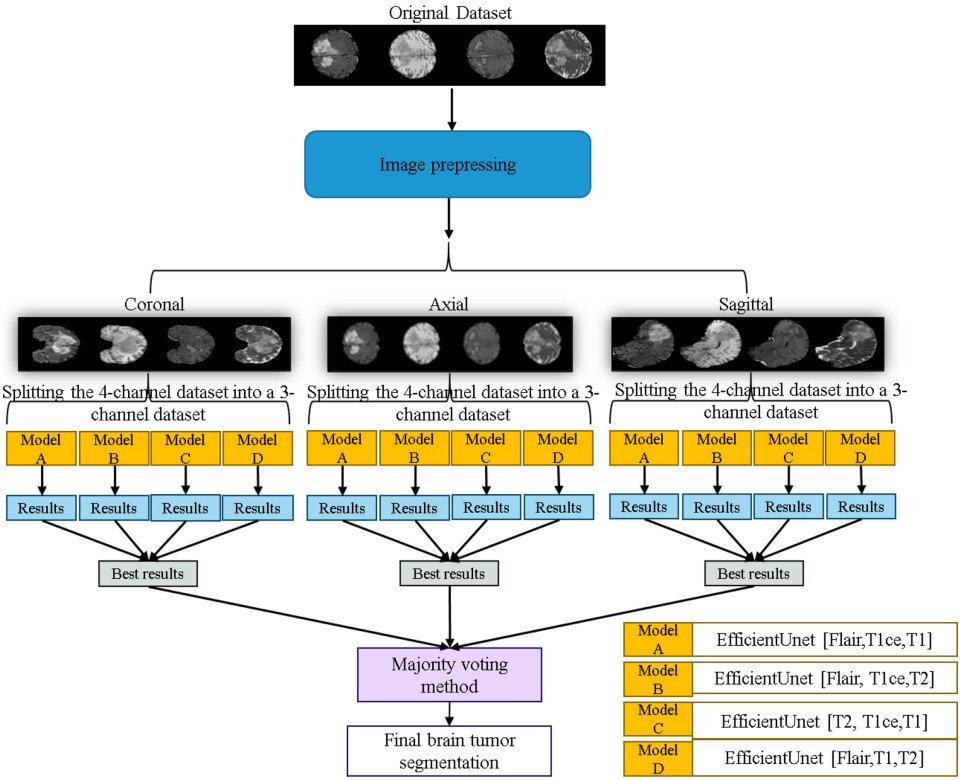

**Figure 11** **Application of the EfficientUNet for brain tumor segmentation workflow.** Original dataset (*Bakas et al., 2017a*; *Bakas et al., 2017b*; *Bakas et al., 2018*; *Clark et al., 2013*; *Menze et al., 2014*, CC0: Public Domain).

selected from these four sets of predictions. Finally, the best predictions from the three orientations are combined using majority voting method (*Van Erp, Vuurpijl & Schomaker, 2002*) to obtain the final predicted segmentation map.

### Performance evaluation metrics

This study adopts accuracy (*Farajzadeh, Sadeghzadeh & Hashemzadeh, 2023*), the generalized Dice loss (GDL) (*Wang et al., 2021*), and the Dice similarity coefficient (DSC) (*Eelbode et al., 2020*) for model evaluation.

Accuracy is a common evaluation metric used in machine learning and deep learning tasks (*Farajzadeh, Sadeghzadeh & Hashemzadeh, 2023*). It measures the overall performance of a model by calculating the ratio of correctly predicted samples to the total number of samples in the dataset, as shown in Eq. (3):

$$accuracy = \frac{Number\ of\ True\ Positives\ (TP) + Number\ of\ True\ Negatives\ (TN)}{Total\ Number\ of\ Samples} \tag{3}$$

where TP represents the cases where the model correctly predicts the positive class, and TN represents the cases where the model correctly predicts the negative class. The accuracy metric provides an overall measure of the model's ability to correctly classify both positive and negative samples. However, in the context of medical image segmentation

tasks, accuracy may not be the most suitable evaluation metric, especially when dealing with highly imbalanced datasets. Instead, evaluation metrics like the GDL and DSC are commonly used to assess the performance of segmentation models.

In the context of MRI brain tumor segmentation, a significant class imbalance poses a major challenge, with 98.46% of the regions belonging to healthy tissue and only 1.54% to brain tumors. To tackle this class imbalance issue, GDL (*Wang et al., 2021*) is employed as the loss function, as shown in Eq. (4):

$$GDL = 1 - 2 * \frac{\left( \sum_i^L w_i \sum_i^i g_{ik} p_{ik} \right)}{\left( \sum_i^L w_i \sum_i^i g_{ik} + p_{ik} \right)} \tag{4}$$

where the total number of labels is denoted as L, and the batch size is represented by k. The weight $w_i$ is set to $1/(\sum_k^i g_{ik})$, where $g_{ik}$ and $p_{ik}$ correspond to the ground truth and predicted images, respectively. GDL is a widely used loss function to address class imbalance, as it measures the similarity between predictions and ground truth labels using the Dice coefficient. A Dice coefficient closer to 1 indicates higher consistency between predicted and ground truth results. By employing GDL as the training loss function, the model can better capture the features of brain tumor regions, leading to improved detection and segmentation capabilities for brain tumors. Moreover, this approach is not influenced by data imbalance issues, effectively narrowing the gap between training samples and evaluation metrics.

Additionally, DSC (*Eelbode et al., 2020*) is used for model evaluation, which is a commonly used metric in brain tumor segmentation. The formula for DSC is as follows:

$$DSC = \frac{2TP}{FN + FP + 2TP} \tag{5}$$

where TP represents true positives, FP denotes false positives, and FN corresponds to false negatives. True positives indicate cases where both the true label and the predicted label are 1, while false positives represent instances where the true label is 1, but the predicted label is 0. Conversely, false negatives indicate situations where the true label is 0, but the predicted label is 1. DSC primarily measures the overlapping area between the predicted lesion region and the actual ground truth region. The Dice similarity coefficient ranges from 0 to 1, where 0 indicates no similarity, and 1 indicates a perfect match. Higher values of the Dice similarity coefficient signify a higher level of consistency between the predicted segmentation result and the true segmentation result.

## RESULTS

The development environment used in this study is Google Colab, which utilizes the Python programming language along with the TensorFlow deep learning framework for development. Within the Google Colab environment, the GPU used is NVIDIA T4, which has a memory capacity of 16 GB GDDR6. This GPU provides additional computational resources, aiding in accelerating the model training and inference processes.

In this section, a comprehensive analysis and comparison of the performance of the EfficientUNet and U-Net models proposed in the 'Methods' section are conducted.

**Table 4  Hyperparameter settings in this study.**

| Parameters | value |
| --- | --- |
| Learning rate | 0.0001 |
| Optimizer | Adam |
| Activation function | Sigmoid |
| Epochs | 30 |

Additionally, the impact of using EfficientNetB4 and EfficientNetV2S as U-Net encoders is explored to investigate their influence on model performance. Furthermore, the prediction systems of EfficientUNet and U-Net are tested. The dataset is sliced into axial, sagittal, and coronal planes, and these slices serve as inputs for prediction. For EfficientUNet, different channel combinations of data are experimented with, and the best results are selected. Ultimately, a majority voting approach is utilized to obtain the final predicted segmentation map. Through these in-depth analyses and tests, the goal of this study is to gain a comprehensive understanding of the performance differences between EfficientUNet and U-Net. Additionally, the study aims to assess how the choice between EfficientNetB4 and EfficientNetV2S as U-Net encoders impacts model performance. This will help in selecting the most suitable model configuration for brain tumor segmentation tasks.

## Implementation details and hyperparameter settings in this study

In this study, a total of 331 labeled cases were randomly divided into training, validation, and test sets in an approximate ratio of 7:1.5:1.5, as described in the Datasets subsection. Specifically, 233 samples were used for training, 49 samples for validation, and 49 samples for testing. The hyperparameter settings in this study are presented in Table 4. To conserve training resources, the Adam optimizer, known for its faster convergence, was chosen as the optimization algorithm. The Sigmoid activation function was selected for binary classification. The learning rate was set to 0.0001, and the training process spanned 30 epochs.

## Comparison between U-Net and EfficientUNet

This section aims to compare the performance of U-Net and EfficientUNet on the axial, coronal, and sagittal planes. In this study, the encoder used in EfficientUNet is EfficientNetV2S. Firstly, the dataset is sliced into two-dimensional data for each of these three planes, and both U-Net and EfficientUNet are utilized for prediction. Through this comparison, the performance of the two models on different planes can be evaluated. Additionally, the predicted results from the axial, coronal, and sagittal planes are integrated using a majority voting approach to further enhance prediction accuracy. By employing this integration method, we can combine the predictions from multiple planes to obtain a more reliable and accurate final result. Such analysis and integration will help assess the performance differences between U-Net and EfficientUNet on different planes and provide an effective way to achieve more precise brain tumor segmentation results.

**Table 5  Training results of axial, coronal and sagittal planes.**

| Model\plane | Axial | | | Coronal | | | Sagittal | | |
|---|---|---|---|---|---|---|---|---|---|
| | Loss | Accuracy | DSC | Loss | Accuracy | DSC | Loss | Accuracy | DSC |
| U-net | 0.0537 | 0.9983 | 0.9452 | 0.0568 | 0.9982 | 0.9420 | 0.0546 | 0.9982 | 0.9442 |
| EfficientUNet (FLAIR, T1ce, T1) | 0.0429 | 0.9986 | 0.9563 | 0.0435 | 0.9986 | 0.9558 | 0.0429 | 0.9986 | 0.9565 |
| EfficientUNet (FLAIR, T1ce, T2) | 0.0428 | 0.9986 | 0.9566 | 0.0440 | 0.9986 | 0.9549 | 0.0433 | 0.9986 | 0.9560 |
| EfficientUNet (FLAIR, T1, T2) | 0.0447 | 0.9985 | 0.9542 | 0.0442 | 0.9986 | 0.9547 | 0.0427 | 0.9986 | 0.9562 |
| EfficientUNet (T2, T1ce, T1) | 0.0531 | 0.9983 | 0.9448 | 0.0516 | 0.9983 | 0.9473 | 0.0508 | 0.9984 | 0.9475 |

**Table 6  Validation results of axial, coronal and sagittal planes.**

| Model\plane | Axial | | | Coronal | | | Sagittal | | |
|---|---|---|---|---|---|---|---|---|---|
| | Loss | Accuracy | DSC | Loss | Accuracy | DSC | Loss | Accuracy | DSC |
| U-net | 0.1088 | 0.9974 | 0.8902 | 0.1299 | 0.9971 | 0.8692 | 0.1197 | 0.9973 | 0.8791 |
| EfficientUNet (FLAIR, T1ce, T1) | 0.1056 | 0.9974 | 0.8937 | 0.1020 | 0.9974 | 0.8971 | 0.1067 | 0.9974 | 0.8925 |
| EfficientUNet (FLAIR, T1ce, T2) | 0.0926 | 0.9975 | 0.9168 | 0.1077 | 0.9973 | 0.8917 | 0.1002 | 0.9975 | 0.8991 |
| EfficientUNet (FLAIR, T1, T2) | 0.1140 | 0.9973 | 0.8849 | 0.1059 | 0.9974 | 0.8931 | 0.1135 | 0.9972 | 0.8852 |
| EfficientUNet (T2, T1ce, T1) | 0.1749 | 0.9963 | 0.8231 | 0.1878 | 0.9959 | 0.8101 | 0.1821 | 0.9948 | 0.8155 |

### Training, validation and testing results on axial plane

According to the results in Table 5, for the training results on the axial plane, EfficientUNet with combinations of (FLAIR, T1ce, T1), (FLAIR, T1ce, T2), and (FLAIR, T1, T2) outperforms U-Net. In particular, the EfficientUNet model with the combination of (FLAIR, T1ce, T2) demonstrates the best performance. In terms of the validation results on the axial plane, according to the results in Table 6, EfficientUNet with combinations of (FLAIR, T1ce, T1), (FLAIR, T1ce, T2) outperforms U-Net. Specifically, the EfficientUNet model with the combination of (FLAIR, T1ce, T2) demonstrates the best performance.In this study, loss curve plots were generated based on the model's training and validation results on axial plane. Figures 12A, 12D display the loss curves during the training process and on the validation set, respectively. From the results in Fig. 12A, it can be observed that both U-Net and EfficientUNet exhibit relatively smooth loss curves during training on the axial plane. However, according to the results in Fig. 12D, the EfficientUNet model with the combination (T2, T1ce, T1) performs poorly on the validation set in the axial plane, while the U-Net model shows significant fluctuations in the validation loss curve on the axial plane. Considering the above observations, it can be concluded that EfficientUNet with combinations such as (FLAIR, T1ce, T1), (FLAIR, T1ce, T2), and (FLAIR, T1ce, T1) outperforms the original U-Net on the validation set in the axial plane. According to the testing results on the axial plane in Table 7, EfficientUNet with combinations such as (FLAIR, T1ce, T1) and (FLAIR, T1ce, T2) outperforms the original U-Net. Among them, EfficientUNet with the combination (FLAIR, T1ce, T1) achieves the best performance. In this study, two test cases, Case A and Case B, were used as axial plane prediction data (Fig. 13A). They were fed into five different models: U-Net, EfficientUNet with combinations (FLAIR, T1ce, T1), (FLAIR, T1ce, T2), (FLAIR, T1ce, T2), and (FLAIR, T1,

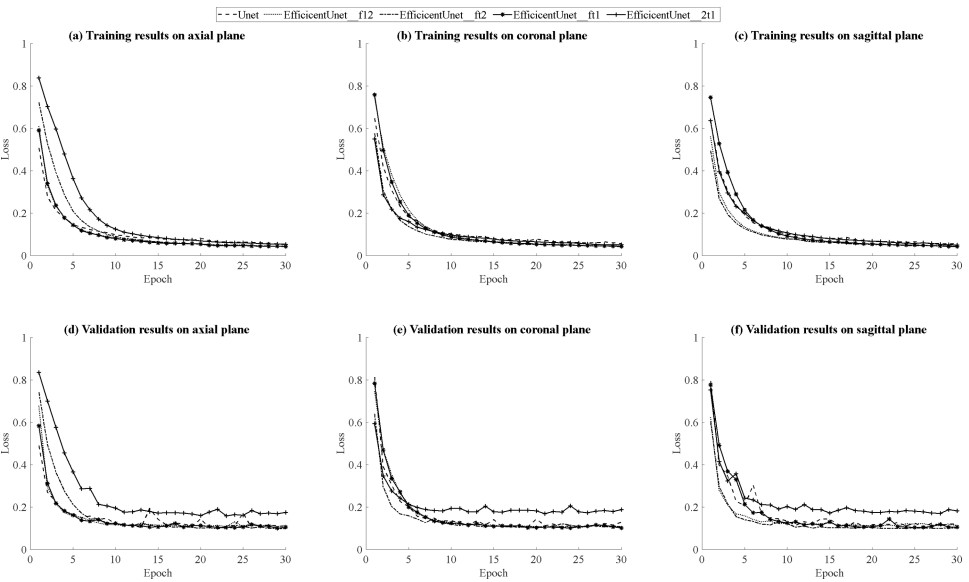

**Figure 12** (A–F) Loss curves during the training process and on the validation set on three plans.

**Table 7** Testing results of axial, coronal and sagittal planes.

| Model\plane | Axial | | | Coronal | | | Sagittal | | |
|---|---|---|---|---|---|---|---|---|---|
| | Loss | Accuracy | DSC | Loss | Accuracy | DSC | Loss | Accuracy | DSC |
| U-net | 0.1022 | 0.9973 | 0.8966 | 0.1037 | 0.9973 | 0.8952 | 0.1066 | 0.9973 | 0.8924 |
| EfficientUNet (FLAIR, T1ce, T1) | 0.0946 | 0.9975 | 0.9046 | 0.1015 | 0.9973 | 0.8968 | 0.1064 | 0.9974 | 0.8927 |
| EfficientUNet (FLAIR, T1ce, T2) | 0.0962 | 0.9975 | 0.9033 | 0.0960 | 0.9972 | 0.9035 | 0.1046 | 0.9974 | 0.8948 |
| EfficientUNet (FLAIR, T1, T2) | 0.1035 | 0.9973 | 0.8955 | 0.1077 | 0.9974 | 0.8912 | 0.1111 | 0.9973 | 0.8879 |
| EfficientUNet (T2, T1ce, T1) | 0.1461 | 0.9963 | 0.8519 | 0.1479 | 0.9961 | 0.8505 | 0.1540 | 0.9948 | 0.8440 |

T2) for segmentation prediction. Figure 13B illustrates the segmentation results of these five models, where white represents correctly segmented regions, and orange indicates incorrectly segmented regions. Based on the observation of Case A, it can be seen that the EfficientUNet (FLAIR, T1ce, T2) model performs the best, achieving the highest Dice similarity coefficient of 0.9684. In contrast, for Case B, the EfficientUNet (T2, T1ce, T1) model predicts more inaccuracies compared to other models, with the lowest Dice similarity coefficient of 0.8886. The EfficientUNet (FLAIR, T1ce, T1) model exhibits the best performance in Case B, achieving the highest Dice similarity coefficient of 0.9772.

### Training, validation and testing results on coronal plane

The training results of U-Net and EfficientUNet on the coronal plane are shown in Table 5. According to the data in the table, EfficientUNet in combination with (FLAIR, T1ce, T1), (FLAIR, T1ce, T2), (FLAIR, T1, T2), and (T2, T1ce, T1) outperforms the original U-Net on the coronal plane. Among these combinations, EfficientUNet with (FLAIR, T1ce, T1) exhibits the best performance. The validation results of U-Net and EfficientUNet

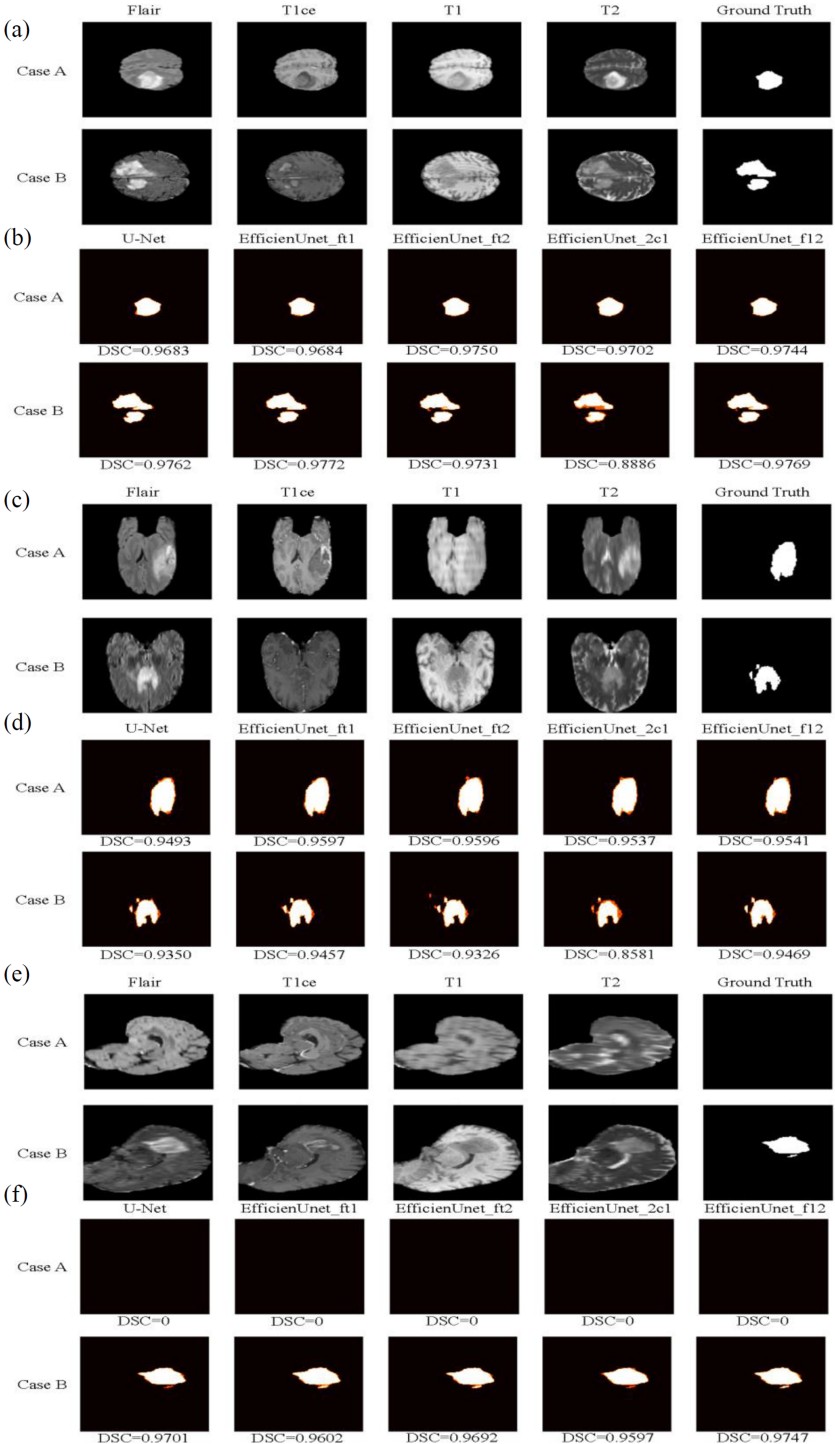

**Figure 13 The brain tumor segmentation results of five different models on two sample cases, Case A and Case B.** Original Dataset (*Bakas et al., 2017a*; *Bakas et al., 2017b*; *Bakas et al., 2018*; *Clark et al., 2013*; *Menze et al., 2014*, CC0: Public Domain). (A) and (B) depict the prediction data and segmentation results of these five models on the axial plane. (C) and (D) display the prediction data and segmentation results on the coronal plane, while (E) and (F) exhibit the prediction data and segmentation results on the sagittal plane.

on the coronal plane are shown in Table 6. EfficientUNet in combination with (FLAIR, T1ce, T1), (FLAIR, T1ce, T2), and (FLAIR, T1, T2) outperforms U-Net in terms of performance. Among these combinations, EfficientUNet with (FLAIR, T1ce, T1) exhibits the best performance. Based on the model training and validation results on coronal plane, the curves were organized and presented in Figs. 12B, 12E displaying the training loss curve and validation loss curve. From the results shown in Fig. 12B, both U-Net and EfficientUNet exhibit relatively smooth loss curves during training on the coronal plane. However, according to the results in Fig. 12E, the EfficientUNet model with (T2, T1ce, T1) combination performs poorly on the validation set for the coronal plane, while U-Net shows larger fluctuations in the validation loss curve. Considering these observations, it can be concluded that EfficientUNet in combination with (FLAIR, T1ce, T1), (FLAIR, T1ce, T2), and (FLAIR, T1ce, T1) outperforms the original U-Net in terms of validation performance on the coronal plane. The test results of U-Net and EfficientUNet on the coronal plane are shown in Table 7. From Table 7, it can be observed that EfficientUNet in combination with (FLAIR, T1ce, T1) and (FLAIR, T1ce, T2) outperforms the original U-Net in the axial plane test. In particular, the combination of EfficientUNet with (FLAIR, T1ce, T2) has proven to be the most outstanding model.

In this study, two sample cases, Case A and Case B, were used as coronal plane prediction data (Fig. 13C). These cases were separately fed into U-Net and EfficientUNet with different combinations: (FLAIR, T1ce, T1), (FLAIR, T1ce, T2), (FLAIR, T1ce, T2), and (FLAIR, T1, T2), resulting in a total of 5 models for segmentation prediction. Figure 13D illustrates the segmentation results of U-Net and EfficientUNet with (FLAIR, T1ce, T1), (FLAIR, T1ce, T2), (FLAIR, T1ce, T2), and (FLAIR, T1, T2) on the coronal plane. In Fig. 13, the white regions represent correctly segmented parts, while the orange regions represent incorrectly segmented parts. Upon analyzing Case A, it can be observed that the EfficientUNet model with (FLAIR, T1ce, T1) yields the best performance, with a Dice similarity coefficient of 0.9597. On the other hand, in Case B, the EfficientUNet model with (T2, T1ce, T1) predicts more inaccurately, with the lowest Dice similarity coefficient of 0.8581, while the EfficientUNet model with (FLAIR, T1, T2) achieves the highest Dice similarity coefficient of 0.9469, outperforming other models. These results suggest that EfficientUNet with combinations such as (FLAIR, T1ce, T1), (FLAIR, T1ce, T2), and (FLAIR, T1, T2) performs favorably in the segmentation results compared to the original U-Net on the coronal plane.

### Training, validation and testing results on sagittal plane

The training results of U-Net and EfficientUNet on the sagittal plane are shown in Table 5. According to the data in Table 7, EfficientUNet with combinations such as (FLAIR, T1ce, T1), (FLAIR, T1ce, T2), (FLAIR, T1, T2), and (T2, T1ce, T1) outperform the original U-Net in terms of training performance on the sagittal plane. Among these combinations, EfficientUNet with (FLAIR, T1, T2) stands out as the best-performing model. The validation results of U-Net and EfficientUNet on the sagittal plane are presented in Table 6. According to the data in Table 6, EfficientUNet with combinations such as (FLAIR, T1ce, T1), (FLAIR, T1ce, T2), and (FLAIR, T1, T2) outperforms the original U-Net in terms of validation performance on the sagittal plane. Among these combinations, EfficientUNet with (FLAIR,

T1ce, T2) stands out as the best-performing model. The results of model training and validation on on sagittal plane were summarized into line charts, including loss curves and validation loss curves, as shown in Figs. 12C, 12F. According to the results in Fig. 12C, both U-Net and EfficientUNet exhibit relatively smooth loss curves during sagittal plane training. However, based on the results in Fig. 12F, it can be observed that EfficientUNet with the combination (T2, T1ce, T1) shows less favorable performance on the sagittal plane validation set, with more significant fluctuations in the validation loss curve compared to other models and U-Net. In conclusion, EfficientUNet with combinations like (FLAIR, T1ce, T1), (FLAIR, T1ce, T2), and (FLAIR, T1ce, T1) outperforms the original U-Net in terms of sagittal plane validation performance. The test results of U-Net and EfficientUNet on the sagittal plane are shown in Table 7. According to Table 7, EfficientUNet with combinations like (FLAIR, T1ce, T2) and (FLAIR, T1ce, T1) outperforms the original U-Net in terms of sagittal plane testing performance. In particular, the combination of EfficientUNet with (FLAIR, T1ce, T2) has been proven to be the most outstanding model.

Using two sample cases, Case A and Case B, as sagittal plane prediction data (Fig. 13E), they were separately input into U-Net and EfficientUNet with combinations such as (FLAIR, T1ce, T1), (FLAIR, T1ce, T2), (FLAIR, T1ce, T2), and (FLAIR, T1, T2), to perform segmentation prediction. Figure 13F shows the segmentation results of U-Net and EfficientUNet with (FLAIR, T1ce, T1), (FLAIR, T1ce, T2), (FLAIR, T1ce, T2), and (FLAIR, T1, 2), as well as the original U-Net. The white regions in Fig. 13 represent correct segmentation, while the orange regions represent incorrect segmentation. In Case A, it can be observed that all five models, U-Net and EfficientUNet with various combinations, make accurate predictions. In Case B, it can be observed that EfficientUNet with (FLAIR, T1, T2) performs the best, with the highest Dice similarity coefficient of 0.9747.

Based on the above model comparisons, it can be observed that EfficientUNet with (FLAIR, T1ce, T2) performs the best in the axial plane dataset, EfficientUNet with (FLAIR, T1ce, T1) performs the best in the coronal plane dataset, and EfficientUNet with (FLAIR, T1, T2) performs the best in the sagittal plane dataset.

## Comparison between EfficientNetB4 and EfficientNetV2S

This section compares the performance differences of EfficientNetB4 and EfficientNetV2S as U-Net encoders on training and validation sets, as well as the performance differences after adding attention gates and residual layers to the decoders on training and validation sets. The relevant results are presented in Tables 8 and 9. In the case of EfficientNetB4 with the addition of attention gates in the decoder, this corresponds to the EAR-U-net architecture proposed by *Wang et al. (2021)*. The inclusion of "EfficientNetB4" and the "ag" label in Tables 8 and 9 indicates the presence of the EAR-U-net architecture.

In the training and validation dataset, based on different combinations of data sets such as (FLAIR, T1, T2), (FLAIR, T1ce, T2), and (T2, T1ce, T1), EfficientNetV2S as the U-Net encoder demonstrates lower loss functions, higher accuracy, and Dice coefficient scores compared to EfficientNetB4. Furthermore, EfficientNetV2S also requires less time for training steps when compared to EfficientNetB4. Only in the (FLAIR, T1ce, T1) dataset combination, EfficientNetV2S performs equivalently in terms of loss functions, accuracy,

**Table 8** **Comparison of EfficientNetB4 and EfficientNetV2S on the training set.** f12 corresponds to the (FLAIR, T1, T2) dataset combination, ft2 corresponds to the (FLAIR, T1ce, T2) dataset combination, ft1 corresponds to the (FLAIR, T1ce, T2) dataset combination, 2t1 corresponds to the (T2, T1ce, T1) dataset combination, and EfficientNetB4 with the inclusion of "ag" label represents the EAR-U-net architecture (*Wang et al., 2021*).

| Model | Loss | Accuracy | DSC | Time (1 step) | GPU ram |
|---|---|---|---|---|---|
| EfficientNetB4_f12 | 0.0461 | 0.9985 | 0.9528 | 737 ms | 14.2 GB |
| EfficientNetB4_f12_ag | 0.0445 | 0.9985 | 0.9545 | 796 ms | 14.2 GB |
| EfficientNetV2S_f12 | 0.0447 | 0.9986 | 0.9542 | 530 ms | 8.8 GB |
| EfficientNetV2S_f12_ag | 0.0457 | 0.9985 | 0.9530 | 622 ms | 8.8 GB |
| EfficientNetB4_ft2 | 0.0437 | 0.9985 | 0.9556 | 700 ms | 14.2 GB |
| EfficientNetB4_ft2_ag | 0.0455 | 0.9985 | 0.9533 | 796 ms | 14.2 GB |
| EfficientNetV2S_ft2 | 0.0428 | 0.9986 | 0.9566 | 550 ms | 8.8 GB |
| EfficientNetV2S_ft2_ag | 0.0440 | 0.9985 | 0.9553 | 615 ms | 8.8 GB |
| EfficientNetB4_ft1 | 0.0429 | 0.9986 | 0.9563 | 716 ms | 14.2 GB |
| EfficientNetB4_ft1_ag | 0.0417 | 0.9986 | 0.9572 | 796 ms | 14.2 GB |
| EfficientNetV2S_ft1 | 0.0429 | 0.9986 | 0.9563 | 512 ms | 8.8 GB |
| EfficientNetV2S_ft1_ag | 0.0440 | 0.9985 | 0.9553 | 615 ms | 8.8 GB |
| EfficientNetB4_2c1 | 0.0548 | 0.9982 | 0.9431 | 759 ms | 14.2 GB |
| EfficientNetB4_2c1_ag | 0.0534 | 0.9983 | 0.9437 | 802 ms | 14.2 GB |
| EfficientNetV2S_2c1 | 0.0531 | 0.9983 | 0.9448 | 605 ms | 8.8 GB |
| EfficientNetB4_2c1_ag | 0.0511 | 0.9984 | 0.9463 | 623 ms | 8.8 GB |

and Dice coefficient scores to EfficientNetB4, but with a shorter required training time. Additionally, during model training, the GPU memory usage for EfficientNetV2S is 14.2GB, which is lower than the 17.3GB used by EfficientNetB4. After incorporating attention gates and residual layers to the decoders, models such as EfficientNetB4 (FLAIR, T1, T2), EfficientNetB4 (FLAIR, T1ce, T1), EfficientNetB4 (FLAIR, T1, T2), and EfficientNetV2S (FLAIR, T1, T2) exhibit improved loss functions, accuracy, and Dice coefficient scores during training compared to the original models without these additions. However, in the validation set, the performance of all models with the addition of attention gates and residual layers in the decoder was unsatisfactory. Furthermore, the incorporation of attention gates and residual layers also led to an increase in model training time.

Based on the outcomes of model training and validation, the data has been synthesized into graphical representations, including curve graphs and validation loss curve graphs, presented in Fig. 14. Across all dataset combinations within the training set, EfficientNetB4 demonstrates swifter convergence, while EfficientNetV2S showcases superior final loss function performance. Furthermore, when utilizing EfficientNetV2S as the encoder, models featuring the inclusion of attention gates and residual layers in the decoder exhibit accelerated convergence during training compared to their initial forms. Conversely, when EfficientNetB4 serves as the encoder, models with added attention gates and residual layers in the decoder tend to manifest a slower convergence rate during training. Notably, within the (T2, T1ce, T1) dataset, EfficientNetB4 exhibits reduced stability during training. In the context of the validation set, encompassing three dataset combinations: (FLAIR, T1, T2),

**Table 9  Comparison of EfficientNetB4 and EfficientNetV2S on the validation set.** f12 corresponds to the (FLAIR, T1, T2) dataset combination, ft2 corresponds to the (FLAIR, T1ce, T2) dataset combination, ft1 corresponds to the (FLAIR, T1ce, T2) dataset combination, 2t1 corresponds to the (T2, T1ce, T1) dataset combination, and EfficientNetB4 with the inclusion of the "ag" label represents the EAR-U-net architecture (*Wang et al., 2021*).

| Model | Loss | Accuracy | DSC | Time (1 step) | GPU ram |
|---|---|---|---|---|---|
| EfficientNetB4_f12 | 0.1151 | 0.9973 | 0.8837 | 737 ms | 14.2 GB |
| EfficientNetB4_f12_ag | 0.1155 | 0.9972 | 0.8830 | 796 ms | 14.2 GB |
| EfficientNetV2S_f12_ | 0.1140 | 0.9973 | 0.8849 | 530 ms | 8.8 GB |
| EfficientNetV2S_f12_ag | 0.1154 | 0.9973 | 0.8831 | 622 ms | 8.8 GB |
| EfficientNetB4_ft2 | 0.1046 | 0.9974 | 0.8948 | 700 ms | 14.2 GB |
| EfficientNetB4_ft2_ag | 0.1172 | 0.9974 | 0.8802 | 796 ms | 14.2 GB |
| EfficientNetV2S_ft2 | 0.0926 | 0.9975 | 0.9168 | 550 ms | 8.8 GB |
| EfficientNetV2S_ft2_ag | 0.1059 | 0.9973 | 0.8931 | 615 ms | 8.8 GB |
| EfficientNetB4_ft1 | 0.1056 | 0.9973 | 0.8937 | 716 ms | 14.2 GB |
| EfficientNetB4_ft1_ag | 0.1399 | 0.9968 | 0.8483 | 796 ms | 14.2 GB |
| EfficientNetV2S_ft1 | 0.1056 | 0.9973 | 0.8937 | 512 ms | 8.8 GB |
| EfficientNetV2S_ft1_ag | 0.1062 | 0.9973 | 0.8927 | 615 ms | 8.8 GB |
| EfficientNetB4_2c1 | 0.1772 | 0.9964 | 0.8205 | 759 ms | 14.2 GB |
| EfficientNetB4_2c1_ag | 0.1782 | 0.9964 | 0.8195 | 802 ms | 14.2 GB |
| EfficientNetV2S_2c1 | 0.1749 | 0.9965 | 0.8231 | 605 ms | 8.8 GB |
| EfficientNetV2S_2c1_ag | 0.1818 | 0.9963 | 0.8081 | 623 ms | 8.8 GB |

(FLAIR, T1ce, T1), and (FLAIR, T1ce, T2), EfficientNetV2S consistently displays swifter convergence relative to EfficientNetB4, except in the case of the (FLAIR, T1ce, T2) dataset combination where convergence is comparatively slower. Nonetheless, across all dataset combinations, EfficientNetV2S consistently demonstrates a more favorable final loss function performance compared to EfficientNetB4.Furthermore, when EfficientNetV2S is employed as the encoder, models incorporating attention gates and residual layers in the decoder exhibit hastened convergence during training, yet these models also show a slightly reduced degree of stability during the training process. Conversely, when EfficientNetB4 is the encoder, models augmented with attention gates and residual layers in the decoder tend to experience a slower pace of convergence during training. Additionally, within the (T2, T1ce, T1) dataset, EfficientNetB4 demonstrates signs of instability during training. Considering all the aforementioned comparisons, EfficientNetV2S outperforms EfficientNetB4 across various parameters, encompassing loss function, accuracy, Dice coefficient score, training time, and GPU memory usage. Although the addition of attention gates and residual layers in the decoder can lead to improved loss function, accuracy, and Dice coefficient scores during training with specific dataset combinations, it ultimately results in reduced performance in these aspects during validation across all models. Furthermore, the inclusion of attention gates and residual layers leads to an increase in model training time. Therefore, selecting EfficientNetV2S as the U-Net encoder without the integration of attention gates and residual layers is a prudent choice, given its lightweight and efficient performance characteristics.

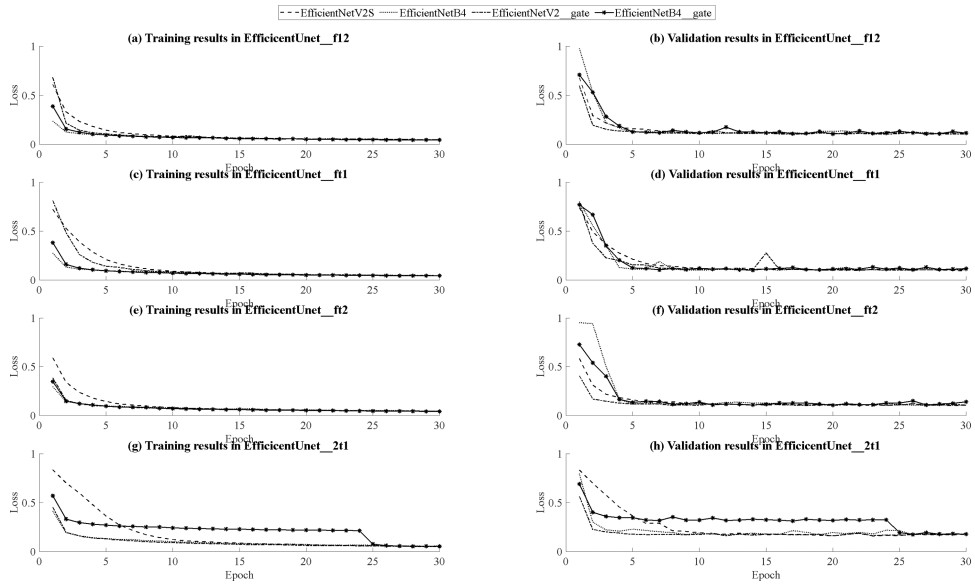

**Figure 14** **Comparison graphs of training and validation loss curves for EfficientNetB4 and Efficient-NetV2S.** (A, B) Training and validation curve comparison for EfficientUNet_f12, (C, D) Training and validation curve comparison for EfficientUNet_ft1, (E, F) Training and validation curve comparison for EfficientUNet_ft2, (G, H) Training and validation curve comparison for EfficientUNet_2t1.

## Applying majority voting method in U-net and EfficientUNet

This study will discuss the application of majority voting methods to both U-net and EfficientUNet models, and compare the differences in segmentation effectiveness when employing majority voting for the axial, coronal, and sagittal planes. Majority voting, as the name implies, involves selecting the final result through a majority decision among the predictions from 2n+1 models. This approach enhances model accuracy by combining predictions from multiple models to achieve more reliable outcomes. Based on the training results in Table 5, the optimal choices for the axial, coronal, and sagittal plane models within the EfficientUNet majority voting approach are EfficientUNet (FLAIR, T1ce, T2), EfficientUNet (FLAIR, T1ce, T1), and EfficientUNet (FLAIR, T1, T2). Table 10 demonstrates that using the majority voting method yields improved performance for axial, coronal, and sagittal plane segmentation for U-net and EfficientUNet models compared to the original models. Furthermore, the performance of EfficientUNet with majority voting method surpasses that of U-net with majority voting method.

Moreover, using the predicted data from Fig. 13, predictions are made using majority voting for both U-net and EfficientUNet, and these are compared against the original models. Figure 15 presents the segmented outcomes of U-net and EfficientUNet with majority voting method, juxtaposed with the original models in Fig. 13. White regions represent correctly segmented portions, while orange areas indicate segmentation errors. In cases A and B, both U-net and EfficientUNet demonstrate improved segmentation compared to the original models when employing the majority voting method. Notably, EfficientUNet with majority voting method outperforms U-net with majority voting

**Table 10   Area of defective MRI images.**

| Model | Loss | Accuracy | DSC |
|---|---|---|---|
| U-net on axial plane | 0.1022 | 0.9973 | 0.8966 |
| U-net on coronal plane | 0.1037 | 0.9973 | 0.8952 |
| U-net on sagittal plane | 0.1066 | 0.9973 | 0.8924 |
| EfficientUNet on axial plane | 0.0962 | 0.9975 | 0.9033 |
| EfficientUNet on coronal plane | 0.1015 | 0.9973 | 0.8968 |
| EfficientUNet on sagittal plane | 0.1111 | 0.9973 | 0.8879 |
| U-net with majority voting method | 0.0902 | 0.9976 | 0.9091 |
| EfficientUNet with majority voting method | 0.0866 | 0.9977 | 0.9128 |

method in terms of segmentation effectiveness. In summary, the findings of this study indicate that utilizing majority voting methods can enhance model performance, with EfficientUNet with majority voting method outperforming U-net with majority voting method in terms of segmentation outcomes.

# CONCLUSION

Through observation of experimental results, this study indeed found that the EfficientUNet model exhibits better segmentation performance and more stable training compared to the original U-net model. The Dice similarity coefficient of the EfficientUNet model is improved by approximately 0.02% compared to U-net. Furthermore, during the prediction phase, employing the majority voting method on the axial, coronal, and sagittal planes enhances the overall accuracy and segmentation performance of the model. The accuracy of the EfficientUNet model is increased from 0.9973 to 0.9976, and the Dice similarity coefficient is elevated from 0.8966 to 0.9091. Similarly, the accuracy of the EfficientUNet model is raised from 0.9975 to 0.9977, and the Dice similarity coefficient is enhanced from 0.9033 to 0.9128.

This technique can be applied in semi-automated medical diagnosis, where doctors can obtain more accurate and reliable image segmentation results using the EfficientUNet model and the majority voting method. This aids in reducing their workload and speeding up the diagnostic process, preventing the loss of optimal treatment opportunities. By improving the accuracy and efficiency of image segmentation, medical professionals can swiftly identify and locate lesions, further enhancing diagnostic and treatment outcomes. The development of this technology holds the potential to bring more convenience and breakthroughs to the medical field, positively impacting the treatment processes for patients. However, it's important to note that this technique serves as an assistant tool for doctors, and the final diagnostic and treatment decisions still require medical judgment and expertise. Thus, when applying this technology, collaboration with doctors and cautious usage are necessary to ensure accuracy and safety of the results.

While our study has made significant strides in advancing the development of efficient and accurate machine learning approaches for AI-based brain tumor segmentation and diagnosis, it is essential to acknowledge certain limitations. These limitations serve as

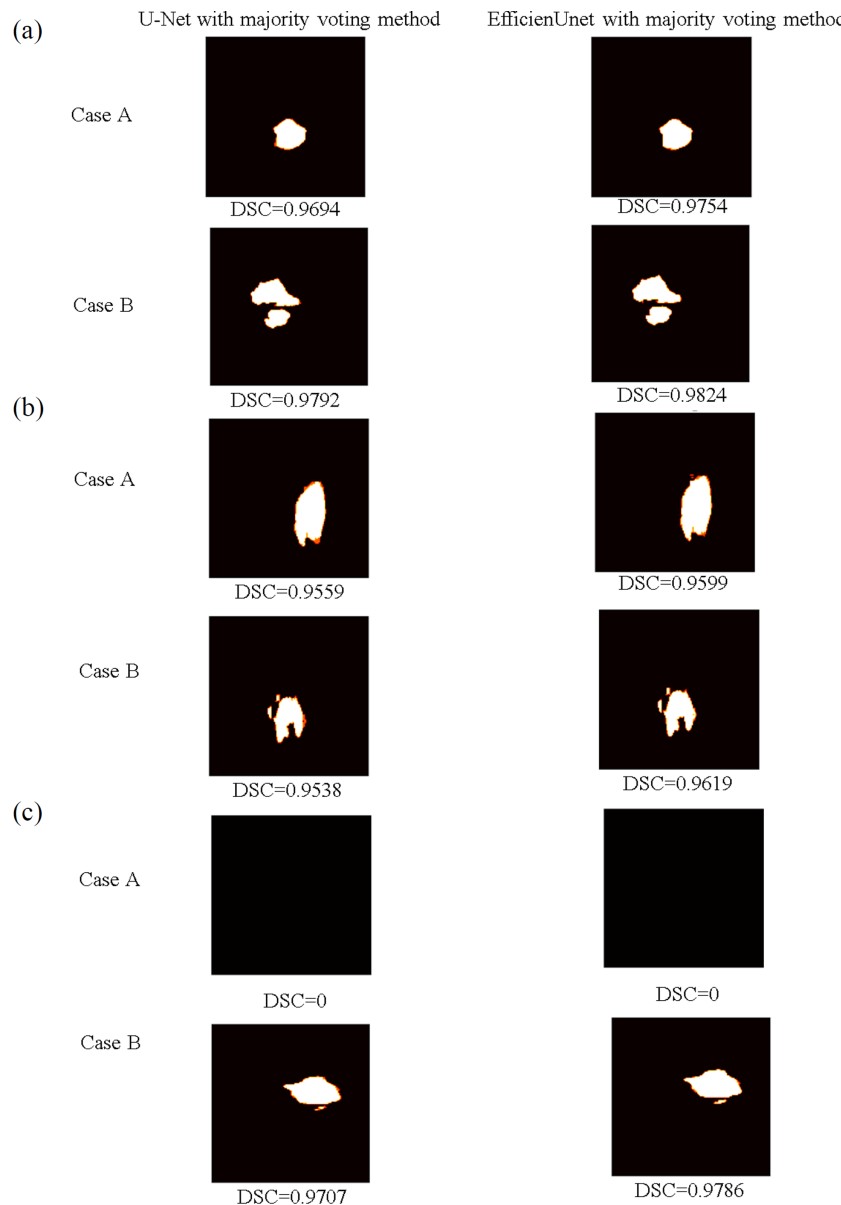

**Figure 15** **The brain tumor segmentation outcomes of U-net and EfficientUNet with the majority voting method are presented for two sample cases, Case A and Case B.** (A) The prediction data and segmentation results on the axial plane are shown. (B) The display the prediction data and segmentation results on the coronal plane. (C) The prediction data and segmentation results on the sagittal plane.

guiding beacons for future research efforts, as we address the following concerns: One limitation of our current approach pertains to the method used for aggregating information from multiple experiments. In this study, we employed the majority voting technique, known for its simplicity and interpretability. However, it may not comprehensively capture the full spectrum of uncertainty in the results. The resource constraints associated with our execution environment on Google Colab, along with the extensive model training

requirements, hindered our ability to perform the required replications and run repeated experiments to report average scores. Future work should explore alternative aggregation methods, such as bootstrap, which can provide a more comprehensive assessment of uncertainty and bolster the robustness of the automated brain tumor segmentation system. Another limitation is associated with our use of a two-dimensional model for brain tumor segmentation. This choice was driven by the need to optimize computational efficiency and practicality, considering the limited resources available on Google Colab. Nonetheless, it may not entirely encapsulate the inherent 3D spatial structure found in medical images. Subsequent research endeavors should delve into the feasibility of integrating three-dimensional models to enhance segmentation accuracy while addressing the computational challenges that accompany 3D data.

In our pursuit of overcoming the identified limitations and propelling the field of AI-based brain tumor segmentation forward, our future work will concentrate on the following pivotal areas:

Alternative aggregation methods: In forthcoming research, we plan to explore alternative approaches for aggregating information from multiple experiments, considering the resource constraints on Google Colab. This will include a thorough investigation of statistical techniques, such as bootstrap, capable of delivering a more comprehensive assessment of uncertainty and model performance. By refining the aggregation process and overcoming resource limitations, our objective is to elevate the reliability and robustness of the automated brain tumor segmentation system.

Three-dimensional models: To surmount the constraints linked to two-dimensional models, especially concerning the representation of 3D spatial structures, our future work will involve the seamless integration of three-dimensional models for brain tumor segmentation. This endeavor will necessitate the development and evaluation of methodologies designed to efficiently handle 3D medical image data, addressing the computational challenges that have been a limitation in the current study. We will explore techniques to make 3D models practical for clinical application while considering the resource limitations of our execution environment. These prospective directions will significantly contribute to the advancement of efficient and accurate AI-based brain tumor segmentation and diagnosis, while effectively addressing the limitations identified in this study.

### Funding
The authors received no funding for this work.

### Competing Interests
The authors declare there are no competing interests.

## Author Contributions

- Shu-You Lin conceived and designed the experiments, performed the experiments, analyzed the data, performed the computation work, prepared figures and/or tables, authored or reviewed drafts of the article, and approved the final draft.
- Chun-Ling Lin conceived and designed the experiments, performed the experiments, prepared figures and/or tables, authored or reviewed drafts of the article, and approved the final draft.

## Data Availability

The code is available in the Supplementary File.

The BraTS datasets are available at Kaggle: https://www.kaggle.com/datasets/aryashah2k/brain-tumor-segmentation-brats-2019.

## Supplemental Information

Supplemental information for this article can be found online at http://dx.doi.org/10.7717/peerj-cs.1754#supplemental-information.

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
