# Peer review of "Brain tumor segmentation using U-Net in conjunction with EfficientNet"

_PeerJ Computer Science, doi:10.7717/peerj-cs.1754_

## Round 0.1 · original submission · Major Revisions

The reviewers have substantial concerns about this manuscript. The authors should provide point-to-point responses to address all the concerns and provide a revised manuscript with the revised parts being marked in different color.

Reviewer 1 ·

Basic reporting

This paper provides an introduction and overview of the importance and challenges of brain tumor segmentation in medical imaging, specifically using Magnetic Resonance Imaging (MRI). The article discusses the limitations of the U-Net model for image segmentation and the benefits of integrating it with EfficientNet and its successor, EfficientNetV2, to improve segmentation results. These improvements are expected to offer medical professionals better tools for early detection and diagnosis of malignant brain tumors, such as Pleomorphic Glioblastoma. However there are some suggestions before publication.

Experimental design

1. **Line 41-43**: While mentioning the "Top Ten Causes of Death Statistics Report", it might be beneficial to provide the year of the report for context.
2. **Line 47-48**: The importance of MRI is mentioned, but the challenges faced by doctors are presented much later.
3. **Line 56-57**: The introduction to U-Net could benefit from a brief summary of its significance before delving into specifics.

Validity of the findings

4. **Line 65-67**: The use of "deep learning architecture" is redundant since U-Net has already been introduced as a deep learning model.
5. **Line 72-73**: Clarify the term "class imbalance."
6. **Line 76-78**: The term "EfficientNet" is suddenly introduced without any precursor.
7. **Line 114-118**: The description of EfficientNetV2 can be more concise.
8. **Line 133**: The "Methods" section typically delves into the specifics of the study. Consider adding a lead-in sentence for clarity.

Cite this review as

Reviewer 2 ·

Basic reporting

The majority of the English writing is good. There are a few places that could be further improved. For example:
line 49 could be changed to “Therefore, automated or semi-automated methods are crucial to facilitate locating and measuring the tumors”.
line 50 could be changed to “Several methods have been proposed for tumor segmentation”
There are some redundant descriptions from line 50 to 55 about ”deep leaning reduces manpower invested in brain tumor segmentation”.
line 69 could be changed to “However, U-Net has some limitations”.
line 74/75 could be changed to “U-Net tends to yield larger model sizes which are computationally expansive”.

The narrative is clear, unambiguous, and professional. But there is some redundancy in it. I suggest the authors delete some repeating descriptions.
The structure is compatible with the PeerJ standards. The figures and tables are relevant and of good quality.

Experimental design

This paper aims at developing efficient and accurate machine learning approaches for AI-based brain tumor segmentation and diagnosis. It integrates the convolutional neural network EfficientNetV2 into the encoders of U-Nets for better performance.
The research problem is well defined and meaningful. This study deepens our understanding of AI-based approaches for brain tumor segmentation.
The authors conducted extensive analysis to address the research problem. In my opinion, most of the analysis is rigorous, professional, and comprehensive. But there are a few places that could be improved.
The methods are described in details. Actually I think the descriptions are a little bit redundant. I suggest the authors keep the keys while removing repeated definitions, concepts, and explanations.
In step 2.2.3, the authors mentioned “using two-dimensional model instead of a three-dimensional model and slicing the three-dimensional data into three orthogonal planes”. I am concerned that this might destroy the latent and inherent structure of the original 3-dimensional images. It would be better that this proposed method can accommodate 3-dimensional models.
The majority voting might not be the optimal way to aggregating informa- tion from multiple experiments. How about bootstrap?

Validity of the findings

The authors covers most relevant literature and introduce the research problem that would advance this field if solved. The connection between this study and existing literature is stated.
In the comparison between UNet and EfficientNet, and the comparison be- tween EfficientNetB4 and EfficientNetV2S, more details needed on replication. Did the authors run repeated experiments and reported the average scores?

Additional comments

I think this study is well proposed and organized. It has some significance and impact. But I found many descriptions are redundant. I suggest the authors make more efforts on the narratives. I recommend a decision “minor revision”.

Annotated reviews are not available for download in order to protect the identity of reviewers who chose to remain anonymous.
Cite this review as

---

## Round 0.2 · accepted · Accept

Reviewers are satisfied with the revisions, and I concur to recommend accepting this manuscript.

Reviewer 1 ·

Basic reporting

After carefully reviewing all the content, because the author has carefully addressed all previous comments, the paper is now ready for publication.

Experimental design

After carefully reviewing all the content, because the author has carefully addressed all previous comments, the paper is now ready for publication.

Validity of the findings

After carefully reviewing all the content, because the author has carefully addressed all previous comments, the paper is now ready for publication.

Cite this review as

Reviewer 2 ·

Basic reporting

The writing of the manuscript has improved. The narrative is unambiguous. Sufficient background/references are provided.

The figures, tables are of fine quality. The authors also give details on the raw data.

The analysis and interpretation are self-contained.

Experimental design

The research problem defined is closed matching the aims and scope of the journal.

The research problem is clearly defined, relevant and meaningful.

The methodology and analysis are logical, suitable, and technical.

The descriptions on dataset, methods, results, and interpretation are in sufficient details.

Validity of the findings

The authors addressed the impact of their findings, which fills a minor gap in the relevant field. The significance and novelty are also stated.

The data has been provided. The analysis is based on statistical ground and follows rigorous procedures.

The conclusions and discussions are well stated, and responding to the research problem defined in the earlier section.

Additional comments

I think the authors have addressed my comments carefully and made suitable changes. Although I still expect further improvement on English writing, I recommend accept this paper.

Cite this review as